# Segmentation Uncertainty Estimation as a Sanity Check for Image Biomarker Studies

**DOI:** 10.3390/cancers14051288

**Published:** 2022-03-02

**Authors:** Ivan Zhovannik, Dennis Bontempi, Alessio Romita, Elisabeth Pfaehler, Sergey Primakov, Andre Dekker, Johan Bussink, Alberto Traverso, René Monshouwer

**Affiliations:** 1Department of Radiation Oncology, Radboud Institute for Health Sciences, Radboud University Medical Center, 6525 GA Nijmegen, The Netherlands; jan.bussink@radboudumc.nl (J.B.); rene.monshouwer@radboudumc.nl (R.M.); 2Department of Radiation Oncology (Maastro), School for Oncology (GROW), Maastricht University Medical Center, 6229 ET Maastricht, The Netherlands; dennis.bontempi@maastro.nl (D.B.); romitaalessio@gmail.com (A.R.); elli.pfaehler@gmail.com (E.P.); andre.dekker@maastro.nl (A.D.); alberto.traverso@maastro.nl (A.T.); 3Department of Radiation Oncology, The Netherlands Cancer Institute, 1066 CX Amsterdam, The Netherlands; 4University Clinic Augsburg, 86156 Augsburg, Germany; 5The D-Lab, Department of Precision Medicine, GROW—School for Oncology, Maastricht University, 6229 ER Maastricht, The Netherlands; s.primakov@maastrichtuniversity.nl

**Keywords:** uncertainty, image biomarkers, radiomics, radiomics harmonization, prognostic modeling

## Abstract

**Simple Summary:**

Radiomics is referred to as quantitative image biomarker analysis. Due to the uncertainty in image acquisition, processing, and segmentation (delineation) protocols, the radiomic biomarkers lack reproducibility. In this manuscript, we show how this protocol-induced uncertainty can drastically reduce prognostic model performance and propose some insights on how to use it for developing better prognostic models.

**Abstract:**

*Problem*. Image biomarker analysis, also known as radiomics, is a tool for tissue characterization and treatment prognosis that relies on routinely acquired clinical images and delineations. Due to the uncertainty in image acquisition, processing, and segmentation (delineation) protocols, radiomics often lack reproducibility. Radiomics harmonization techniques have been proposed as a solution to reduce these sources of uncertainty and/or their influence on the prognostic model performance. A relevant question is how to estimate the protocol-induced uncertainty of a specific image biomarker, what the effect is on the model performance, and how to optimize the model given the uncertainty. *Methods*. Two non-small cell lung cancer (NSCLC) cohorts, composed of 421 and 240 patients, respectively, were used for training and testing. Per patient, a Monte Carlo algorithm was used to generate three hundred synthetic contours with a surface dice tolerance measure of less than 1.18 mm with respect to the original GTV. These contours were subsequently used to derive 104 radiomic features, which were ranked on their relative sensitivity to contour perturbation, expressed in the parameter *η*. The top four (low *η*) and the bottom four (high *η*) features were selected for two models based on the Cox proportional hazards model. To investigate the influence of segmentation uncertainty on the prognostic model, we trained and tested the setup in 5000 augmented realizations (using a Monte Carlo sampling method); the log-rank test was used to assess the stratification performance and stability of segmentation uncertainty. *Results*. Although both low and high *η* setup showed significant testing set log-rank *p*-values (*p* = 0.01) in the original GTV delineations (without segmentation uncertainty introduced), in the model with high uncertainty, to effect ratio, only around 30% of the augmented realizations resulted in model performance with *p* < 0.05 in the test set. In contrast, the low *η* setup performed with a log-rank *p* < 0.05 in 90% of the augmented realizations. Moreover, the high *η* setup classification was uncertain in its predictions for 50% of the subjects in the testing set (for 80% agreement rate), whereas the low *η* setup was uncertain only in 10% of the cases. *Discussion*. Estimating image biomarker model performance based only on the original GTV segmentation, without considering segmentation, uncertainty may be deceiving. The model might result in a significant stratification performance, but can be unstable for delineation variations, which are inherent to manual segmentation. Simulating segmentation uncertainty using the method described allows for more stable image biomarker estimation, selection, and model development. The segmentation uncertainty estimation method described here is universal and can be extended to estimate other protocol uncertainties (such as image acquisition and pre-processing).

## 1. Introduction

### 1.1. Background

Image biomarker analysis, also known as radiomics, is a technique to extract quantitative information (radiomic features or image biomarkers) from routinely acquired medical images, such as computed tomography (CT), positron emission tomography (PET), or magnetic resonance imaging (MRI). Radiomics extraction organically fits clinical routines if annotated data is already present in the clinic. For example, in radiation oncology, where tumors and organs-at-risk are routinely delineated as a part of treatment planning, radiomics can be extracted and stored automatically in the clinical picture archiving and communication system (PACS) [1].

Currently, the image biomarker research focuses either on its direct implementation in prognostic or diagnostic clinical models [2,3] or on image biomarker harmonization and reproducibility [4,5]. Whereas the first research field focuses on the final step of the image biomarker workflow (Figure 1), the latter field works on harmonizing uncertainties induced in the image biomarker values in various stages of Figure 1’s workflow. Each of the workflow steps in Figure 1 could be approached with a range of tools and methods: from purely expert-knowledge-driven hand-crafted to data-driven machine learning approaches.

Recent radiomics studies show that radiomic features lack reproducibility against image acquisition, pre-processing, and segmentation protocols [5]. Therefore, a set of methods referred to as radiomics harmonization is proposed to calibrate radiomic values based on those protocol differences (e.g., scanner signal-to-noise ratio, voxel size, segmentation algorithm, etc.) [4,7,8]. Protocol uncertainty propagates through the Image Biomarker Analysis workflow (Figure 1) and induces uncertainty in image biomarker and prognostic model performance.

A fundamental research question of our manuscript is how to estimate the uncertainty induced by the protocol differences and decide whether a model can benefit from harmonization. We answer this question by defining statistical metrics to estimate image acquisition, pre-processing, or segmentation protocol-induced uncertainty in prognostic modeling. As a specific example, we show how the segmentation uncertainty inherent in clinical cohorts influences model performance.

### 1.2. Problem

Let us start with a concrete example: take into consideration a biomarker (for example, a radiomic feature) given a statistical binary classification problem that has two populations A and B (for instance, short versus long survival stratification). Each population has its population mean (expected value μ) and standard deviation σ, as shown in Figure 2. A biomarker value is sampled from the population distribution and the sample will have its sample mean $\bar{X}$ and sample standard deviation s.

Larger intra-population variance makes two distributions overlap more and, therefore, two populations become less distinguishable. The cause of the variance can be intrinsic (clinical) or due to the measurement (i.e., protocol uncertainty). With the effect, the distance between the groups, and the intra-population variance, an uncertainty measure, matters. One conventional way of quantifying the effect versus uncertainty is Cohen’s *d* (1) effect size measure, defined as a standardized mean difference between groups A and B [9]:(1)$$d_{AB}=\frac{\left|{\bar{X}}_{A}-{\bar{X}}_{B}\right|}{\sqrt{\frac{\left(n_{A}-1\right)s_{A}^{2}+\left(n_{B}-1\right)s_{B}^{2}}{n_{A}+n_{B}-2}}\;}$$

This effect size measure is proportional to the distance $\left|{\bar{X}}_{A}-{\bar{X}}_{B}\right|$ between the groups and is inversely related to the two intra-group variances $s_{A}^{2}$ and $s_{B}^{2}$. This measure, however, does not directly assess uncertainty, induced by the protocol parameters of the workflow shown in Figure 1 for a specific patient, but rather the cumulative effect of both intrinsic (clinical) and measurement (protocol) uncertainties. The latter may also be referred to as intra-patient uncertainty. Measuring the biomarker’s intra-patient uncertainty relative to the effect is crucial, but rarely done due to how the conventional radiomics modeling is performed.

In the conventional process of radiomics modeling, a coupled image-mask pair is introduced to a radiomics extraction software (e.g., pyradiomics [2]). In such an approach, the intra-patient sample size (the number of measurements) equals one (one image-mask pair), which can lead to a poor estimation of a radiomic feature value. In the ideal setting, the intra-patient sample size can be increased by extracting radiomics from several image-mask pairs acquired using different image acquisition (scanner brand, signal-to-noise, reconstruction kernel), segmentation (intra-observer variability), and pre-processing (binning, filtering, resampling) protocols. It is important to mention that *the protocol variation should be adequate for the realistic clinical protocol uncertainty*. The increased sample size then will improve the quality of the estimates (mean biomarker value and its variance). Therefore, an estimate for intra-patient uncertainty in a radiomic feature can be then defined as the total variance (2):(2)$$s_{intrapat}^{2}=s_{acq}^{2}+s_{seg}^{2}+s_{proc}^{2}+2COV_{acq,\;seg}+2COV_{acq,\;proc}+2COV_{\;seg,\;proc}$$
where$s_{a}^{2}$—the intra-patient variance due to protocol ‘*a*’ uncertainty;$COV_{a,b}$—the covariance of protocol ‘*a*’ and ‘*b*’ uncertainties.

This estimate of intra-patient variance can be introduced into an effect size measure derived from (1), in order to measure the ratio between uncertainty and the effect in this specific radiomic feature (3):(3)$$\eta =\frac{E\left[s_{intrapat}\right]}{{\;\Delta }_{effect}}=\frac{E\left[s_{intrapat}\right]}{IQR_{0.25-0.75}\left(X_{i}\right)}$$
where the denominator (effect) can be measured as the interquartile range in the dataset, because whenever the two classes are even in size (e.g., the median stratification is used), the medians of both high- and low-risk groups can be estimated as the dataset’s 0.25 and 0.75 quantiles. The numerator is the estimated value of intra-patient variance for this radiomic feature. If the value of *η* is close to 1, the intra-patient variance is comparable to the effect, which reduces the feature usefulness in the detection of the effect unless harmonization is applied to reduce the uncertainty.

This study was limited to segmentation-induced uncertainty estimation in radiomic values, so a simplified version of Equations (2) and (3) was reduced to Equation (4). The *η* values were estimated with a median interquartile range per patient subject in the numerator (uncertainty estimation) and with the interquartile range of the median estimated per subject in the denominator (effect estimation)
(4)$$\eta =\frac{\tilde{IQR\left(X_{i}\right)}}{IQR\left(\tilde{X_{i}}\right)}$$

## 2. Materials and Methods

### 2.1. Datasets

The two non-small cell lung cancer (NSCLC) cohorts in this study were previously labeled by Aerts et al. as “Lung1” and “Lung2” and utilized in a replication study by Shi et al. [10,11]. The patients in both cohorts were previously treated with (chemo-)radiotherapy and the latest clinical parameters of the cohorts were provided in Shi et al. [11]. For our analysis, 420 Lung1 and 227 Lung2 patient images and respective GTV contours were selected from 422 and 267 records, respectively: the excluded cases contained either no primary tumor contours or missed survival data. The overall survival was used as the outcome. Similarly to the two abovementioned studies, the Lung1 set was used for training, feature ranking and selection, whereas the Lung2 set was used for testing and evaluation. All the DICOM images and RTSTRUCT contour files were converted to SimpleITK-compatible images and binary masks with a spacing of 1 mm^3^ using Plastimatch v 1.9.0 (https://plastimatch.org/, last access on 1 October 2021).

### 2.2. Estimation of Segmentation Uncertainty and the Effect Size of the Radiomic Features

In order to estimate the segmentation uncertainty in radiomic values, we generated binary masks within a clinically relevant surface dice tolerance [12] using a Monte Carlo sampling technique described below. For every patient in the training and testing sets, 300 binary deformed GTV masks were generated using a max surface dice tolerance of 1.18 mm relative to the original contour. The value of 1.18 was taken as a clinically reasonable maximum value following a comprehensive lung tumor (auto-) segmentation study by Primakov et al., where the interobserver variability experiment found the surface dice tolerance of 1.18 mm in the Maastro Interobserver Lung cancer dataset (https://xnat.bmia.nl, last access on 1 October 2021) [13].

Each generated mask was used alongside the reference 3D image to extract radiomic features with 25 HU binning and a 1 mm^3^ voxel size using PyRadiomics v3.0.1 [2]. First-order statistics, shape, gray level co-occurrence (GLCM), gray level run length (GLRLM), gray level dependence (GLDM), gray level size zone (GLSZM), and Neighboring Gray Tone Difference (NGTDM) matrix-based features were extracted—104 radiomic features in total.

The extracted radiomic values for the 300 augmented + one original binary mask for each of 420 subjects in the training set were used to calculate *η* according to Equation (4). The 104 features were then ranked in descending order according to their respective *η* values, as shown in Appendix A.

The Segmentation UnceRtainty Estimation (sure) project code for image and mask processing, radiomics extraction, and subsequent clinical model setup is available online: Segmetation UnceRtainty Estimation repository—https://github.com/Maastro-CDS-Imaging-Group/sure, last access on 1 October 2021.

### 2.3. Prognostic Model Simulation

In order to evaluate segmentation uncertainty and its influence on radiomic values and clinical prognostic models, we selected the top four and bottom four features based on their *η* ranking (Table 1 and Appendix A). For each setup, two identical analysis pipelines were executed based on a Cox proportional hazards model with median stratification similar to the previous studies [2,11,14].

The *pipeline* consisted of the following steps: (1) A Cox proportional hazards model was trained in the Lung1 training set using the four features as the covariates and overall survival as the outcome. (2) Median hazard ratio and model weights were derived based upon the training set and applied in the Lung2 testing set. (3) Performance metrics in the testing set were collected (log-rank statistic and *p*-value). Prior to analysis, all four features were log-scaled and standardized, and the outliers were removed (absolute value > 3σ), similar to Ref. [11], in both augmented (300 + 1 masks per patient) datasets.

In the baseline model, the feature values extracted from the original GTVs were used as the covariates of the model. To measure the pipeline’s sensitivity for segmentation uncertainty, one record was sampled from 301 radiomic values per patient (generated during the Monte Carlo sampling, Figure 3) with replacement 5000 times in both training and testing sets—resulting in a total of 5000 pipeline simulations.

Out of these 5000 pipeline simulations, one was selected based on the least squares of its covariate coefficients and medians of the coefficients; the resulting averaged model was then scored in the abovementioned 5000 realizations of the Lung2 testing set. The 5000 scores were averaged per patient and the final “ensembled” model scores were derived. The general outline of the experiment is shown in Figure 4.

Although *p*-values give a good estimation of statistical performance of both setups, it is clinically relevant to measure how confident the model is in its prediction when segmentation uncertainty is introduced. Using the ensembled Cox model for each patient of the testing set, we measured the agreement of the model by measuring misclassified samples using 301 augmented mask realizations. This was done by scoring the ensembled model in 301 augmented realizations for every subject in the Lung2 testing set according to their stratification mislabeling *δ* (5):(5)$$\delta =\frac{\#MajC-\#MinC}{\#MajC+\#MinC}$$
where*#MajC*—number of samples of the majority class for a subject;*#MinC*—number of samples of the minority class for a subject.

## 3. Results

The original GTV segmentation-based models A and B were performed with a log-rank *p* = 0.011 and 0.012, respectively. After the introduction of segmentation uncertainty in both datasets by sampling 5000 times randomly from the augmented realizations, setup B showed that only 1428 out of 5000 scenarios achieved a log-rank *p* < 0.05 in the test set—a 28.5% chance of getting a significant result while sampling from the clinically adequate range of segmentation uncertainty.

After the ensembled model scoring in test data, a log-rank *p* = 0.1 was achieved in the test set. The setup B performance summary is shown in Figure 5. Setup A, on the other hand, showed robust performance in the testing set’s augmented realizations: its 4508 out of 5000 realizations (90%) showed a log-rank *p* < 0.05 and its ensembled model performed with a log-rank *p* = 0.012, which was similar to the original setup A performance (*p* = 0.011).

Figure 5 shows an integral picture of how segmentation uncertainty influences the statistical performance of both setups A and B. However, it is more illustrative to check a specific realization. Figure 6 shows two sample performances of the same core setup (ensembled model, Figure 5) in two different realizations of the testing set: although exactly the same images of Lung2 subjects were used, using radiomic features that are sensitive for segmentation uncertainty (Setup B) can lead to distinctly worse performance.

In order to measure the agreement of the setup’s scoring different segmentation realizations of the same subject, we scored every subject’s segmentation realization with the ensembled model in both setups. Figure 7 shows that segmentation uncertainty causes setup B to have an 80% agreement only for 50% of patients in the testing set, whereas there is an 80% agreement for 90% of subjects’ setup A.

## 4. Discussion

### 4.1. General Discussion

Image and contour data acquired during diagnostic or treatment planning in the clinic are inherently subject to stochastic (for instance, noise) as well as systematic deviations (different scanners, delineation protocols, analysis methods). When using the data for modeling, we therefore can consider the given dataset as one representation sampled from a broader image protocol uncertainty distribution. An alteration or variability in protocol settings will enlarge the uncertainty in radiomic values, ultimately impacting the image biomarker’s prognostic performance. These protocol variations should be reduced as much as possible, driving the need for radiomics harmonization across the Image BiomarkerAanalysis Workflow (Figure 1). As the uncertainty propagates through the Image Biomarker Analysis Workflow (Figure 1), it influences all subsequent stages of the workflow and, eventually, causes uncertainty in the image biomarker performance. This propagation may cause the terminological ambiguity between the terms “protocol uncertainty”, “protocol-induced uncertainty”, and “image biomarker uncertainty”. We used the first term to describe the variations in the protocol (e.g., variations in delineation), whereas the latter terms were used to describe the variance in the image biomarker performance.

The segmentation uncertainty is an important source of uncertainty in radiomics as, until now, tumor delineation is a manual process in clinical practice and can vary substantially from person to person [13]. In this study, we investigated segmentation uncertainty influence in radiomics and proposed a generalizable method to estimate protocol uncertainty influence in image biomarker-based prognostic models. Even though the framework is described in terms of segmentation uncertainty, the method can be applied to other types of uncertainty. *We would like to strongly emphasize the importance of measuring the uncertainty alongside the performance metrics whenever possible in image biomarker studies*.

In this study, we added three aspects to the uncertainty estimation for image biomarker studies. Firstly, we have added a framework (Equations (2)–(4)) to estimate uncertainty versus effect in radiomic values with regard to a specific parameter(s) variation—in our case—segmentation uncertainty. Although the standard error might be a useful rough estimate for overall uncertainty, it does not measure the influence of a specific protocol parameter. Secondly, we have shown how to model uncertainty propagation through the image biomarker analysis workflow (Figure 1) using Monte Carlo sampling techniques for binary mask sampling and dataset realization aggregation. Thirdly, we showed that with even a simple harmonization technique, the ensembled model with score averaging gives above median performance—the uncertainty estimates can be used to improve model performance.

In our study, we made the assumption that for each patient, the given original contour has uncertainty, and could be contoured slightly differently, for instance, by another radiation oncologist. Figure 5 shows the distribution of performance (expressed as log-rank *p*-value) of 5000 models in different realizations of the testing set—this distribution’s parameters (e.g., the interquartile range or 95% interval) can be used as an estimate of segmentation-induced uncertainty in the image biomarker model. Figure 5 can be considered as an integral picture of Figure 6, where the two sample models’ performances would lead to absolutely different conclusions because of the segmentation uncertainty of max 1.18 mm surface dice tolerance. On the other hand, Figure 7 shows how one model can be uncertain while performing low- and high-risk survival stratification: high uncertainty leads to the model being uncertain in 50% of the subjects (given the mislabeling rate of 20%—Equation (5)).

### 4.2. Limitations and Future Perspective

In our manuscript, we wanted to avoid the topic of evaluating image and binary mask processing’s influence on radiomic values. We tried to keep all the parameters for pre- and post-processing in the pipeline constant by using a standard pipeline so that they do not cause additional uncertainty in data. In other words, we tried to isolate the three sources of uncertainty, namely, acquisition, segmentation, and processing, in order to estimate the individual contribution of each term. One example of image pre-processing influence can be the raw original mask delineation and interpolation. This protocol variation causes different coarseness of the ROI surface.

Image acquisition-uncertainty is hard to directly access in the clinical setting since this requires acquiring a large set of image data samples for each subject, which is not realistic due to time and, in the case of computed tomography, radiation dose constraints. Phantom data lifts the abovementioned constraints and can be used to measure proxy estimates of image acquisition-uncertainty. However, a phantom set does not necessarily represent the complexity and textural patterns of biological tissues, therefore, phantom-estimated image acquisition-uncertainty should be used with precautions [7].

To estimate segmentation uncertainty, we performed 300 binary mask augmentations using the so-called “volume-front” method (see the GitHub repository). However, we expect that other methods for inducing binary mask uncertainty can also be used. For instance, Zwanenburg et al. investigated various contour perturbation techniques and their influence on radiomic values [15]. The augmented radiomic data were distributed normally, therefore, we considered that 300 binary mask augmentations gave a good estimation of segmentation uncertainty in radiomics. Using binary masks with 1 mm^3^ voxel size limits the number of augmented binary masks, as we sample from a finite set of voxels in the uncertainty ring. To mitigate this, segmentation uncertainty might be based on raw RTSTRUCT contour coordinates and not on binary masks [16]. We do not, however, believe that this will fundamentally change the analysis of this paper.

Although in our study we considered every segmentation realization as equally possible (uniformly distributed), this might not be the case in a realistic scenario: a data-driven approach to assign a non-uniform probability to measure a segmentation realization could make an interesting study. In addition, the 5000 number of core setup iterations was selected based on the memory limitation of the computer—more computations could be done, however, we do not expect them to be beneficial, as similar results could be acquired for a much lower number of iterations (~1000).

The proposed generic framework of protocol uncertainty can be used to measure image acquisition, processing, and segmentation uncertainty combined using (2) and (3) to estimate η. This framework is not bound to radiomics only: any model setup that relies on image data with uncertainty can be investigated on its stability versus protocol uncertainty. In this case, instead of radiomic values, the model predicted outcome or embeddings (latent representations) could be benchmarked against protocol uncertainty.

## 5. Conclusions

Image biomarker value estimated only in the original segmentation might be deceiving, as this estimation does not account for segmentation uncertainty. Although a prognostic model might accidentally give a significant stratification performance, this can be a matter of chance given a set of segmentations provided by the clinic. Without segmentation uncertainty introduced and estimated, the model will not be able to accommodate for it. Segmentation uncertainty estimation not only gives clearer insight into model performance but is essential to improve model robustness.

## Figures and Tables

**Figure 1 cancers-14-01288-f001:**
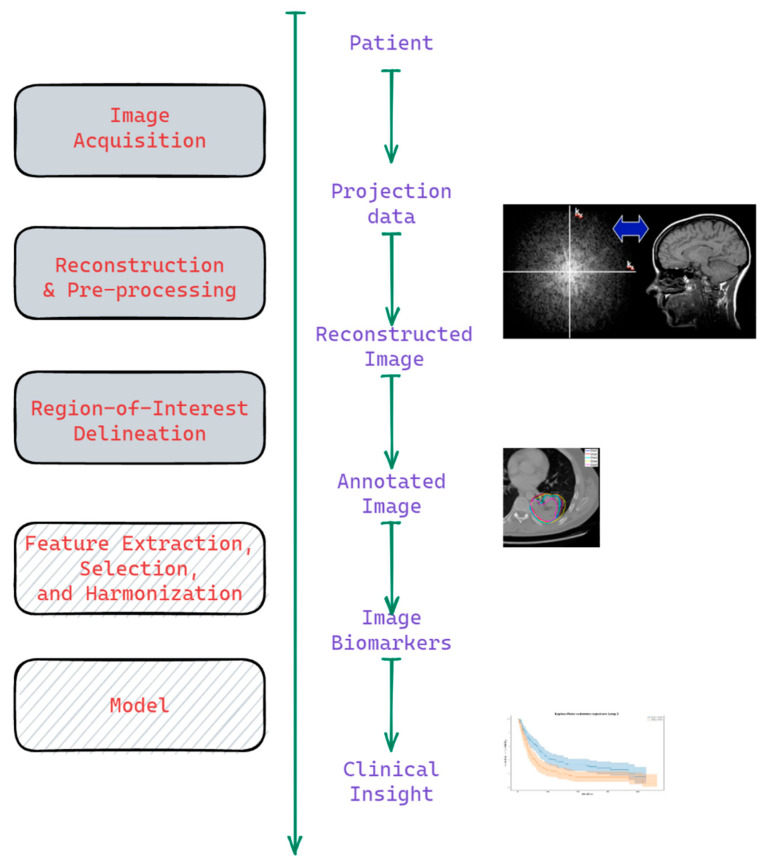
The Workflow of Image Biomarker Analysis: from image acquisition to modeling—each procedure can be performed using a different protocol (set of parameters), thus inducing uncertainty in the image biomarker model performance. Partially adapted from: https://med.stanford.edu/bmrgroup/Research/AcqRecon.html (accessed on 14 January 2022), https://www.nature.com/articles/srep03529 (accessed on 14 January 2022) [6].

**Figure 2 cancers-14-01288-f002:**
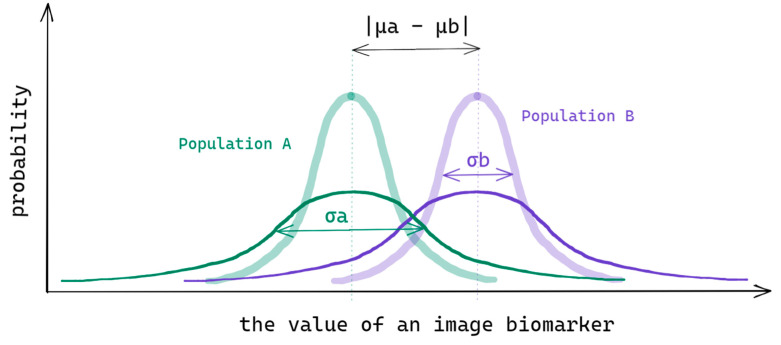
Image biomarker value distributions in populations A and B given relatively low (low opacity curves) and high (full opacity curves) intra-population variance.

**Figure 3 cancers-14-01288-f003:**
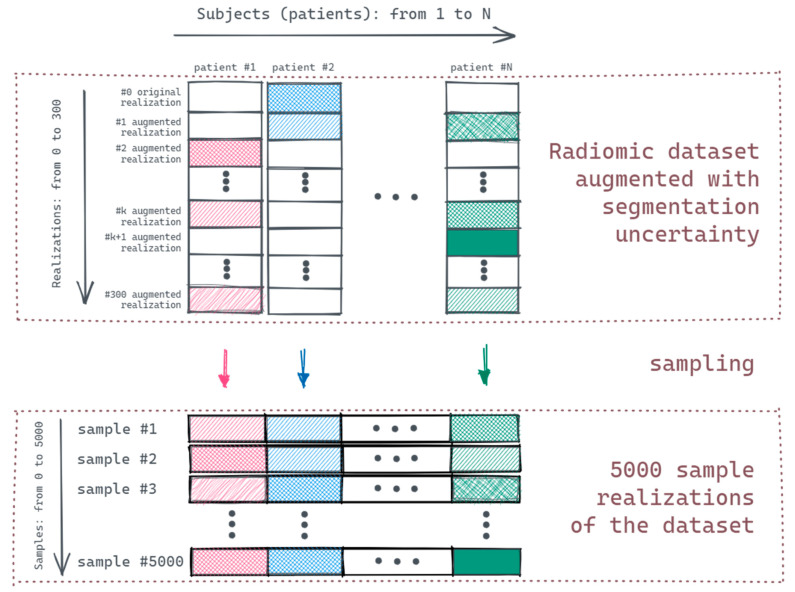
Monte Carlo sampling.

**Figure 4 cancers-14-01288-f004:**
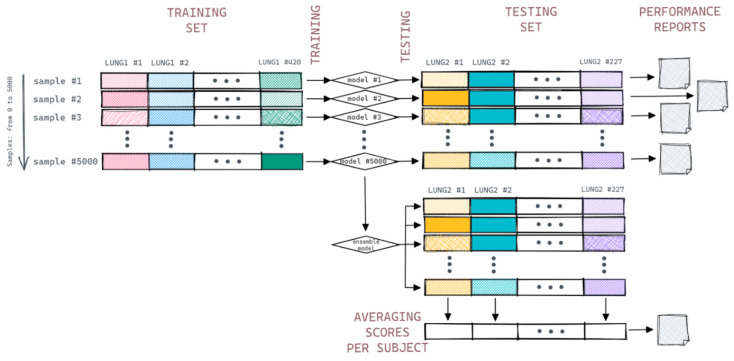
The general outline of the experiment.

**Figure 5 cancers-14-01288-f005:**
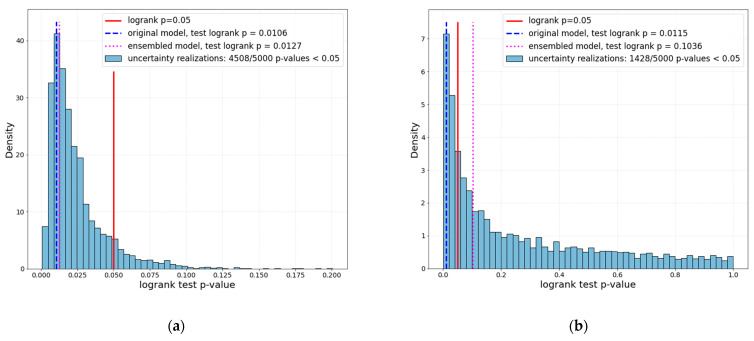
Log-rank *p*-value distribution in the test set for setups A (**a**, low *η*) and B (**b**, high *η*).

**Figure 6 cancers-14-01288-f006:**
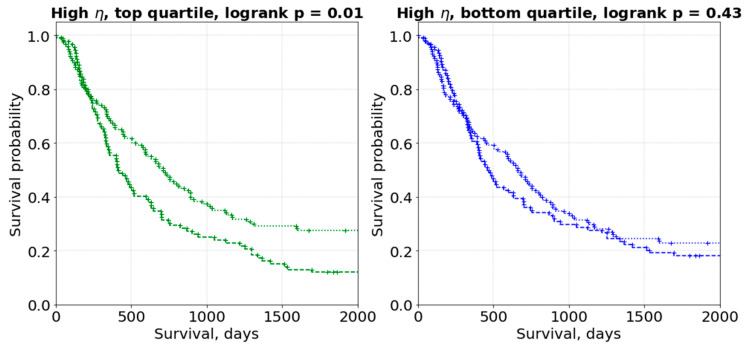
Segmentation uncertainty influence in the high *η* (setup B): two sample testing set realizations result in different performances.

**Figure 7 cancers-14-01288-f007:**
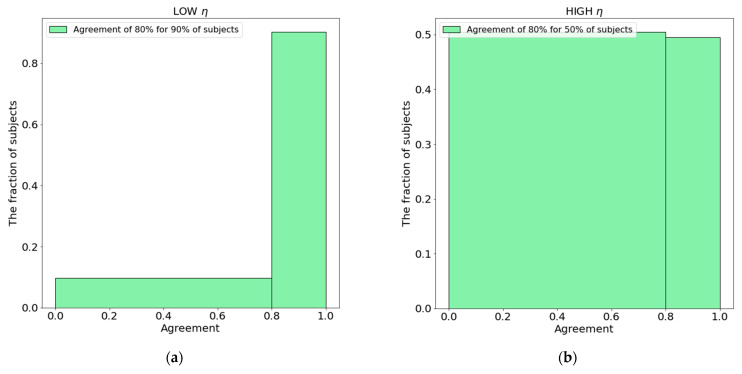
Segmentation uncertainty and patient stratification agreement *δ* Equation(5) in setups A (**a**, low *η*) and B (**b**, high *η*).

**Table 1 cancers-14-01288-t001:** Two sets of radiomic features selected for analysis.

Setup A—Low *η*		Setup B—High *η*	
Feature	*η*	Feature	*η*
first-order Maximum	0.0	glszm GrayLevelVariance	0.2944
gldm GrayLevelNonUniformity	0.0108	glrlm RunEntropy	0.3066
glrlm GrayLevelNonUniformity	0.0115	glcm MCC	0.3216
ngtdm Coarseness	0.0129	glszm GrayLevelNonUniformityNormalized	0.4005

## Data Availability

The Lung1 dataset is available at https://wiki.cancerimagingarchive.net/display/Public/NSCLC-Radiomics (last access 1 October 2021). The code is public and available at https://github.com/Maastro-CDS-Imaging-Group/sure (accessed on 14 January 2022).

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
