# Peer review of "Segmentation Uncertainty Estimation as a Sanity Check for Image Biomarker Studies"

_cancers, 2022, doi:10.3390/cancers14051288_

Round 1
Reviewer 1 Report
As noted by the authors, image biomarker value estimated only in the original segmentation might be deceiving as this estimation does not account for segmentation uncertainty. The main claim of this paper is that segmentation uncertainty estimation does not only give clearer insight into model performance but is essential to improve model robustness.
The authors make the distinction between the terms “protocol uncertainty”, “protocol-induced uncertainty”, and “image biomarker uncertainty”. They use the first term to describe the variations in the protocol, whereas the latter terms are used to describe the variance in the image biomarker performance.
The authors investigate segmentation uncertainty influence in radiomics and propose a generalizable method to estimate protocol uncertainty influence in image biomarker-based prognostic models. They strongly emphasize the importance of measuring the uncertainty alongside the performance metrics whenever possible in image biomarker studies.
They further added three aspects to the uncertainty estimation for image biomarker studies. Firstly, they add a framework to estimate uncertainty versus effect in radiomic values with regard to segmentation uncertainty. Secondly, they show how to model uncertainty propagation through the image biomarker analysis workflow. Thirdly, they show that even a simple harmonization technique can improve model performance.
The limitations of the proposed approach are highlighted in the discussion and the influence on their analysis is mentioned. The authors claim that these limitations would not fundamentally change their conclusions.
Results show how segmentation uncertainty influences statistical performance.
In view of all the above, this reviewer recommends publication of this manuscript.
Author Response
≫We thank the reviewer for the high evaluation of our work!
Reviewer 2 Report
The authors evaluated the system with them we can estimate the uncertainties during the analyses on radionics comparison.
I found the work well done, in terms on scientific approach and on the data presentation.
They are concentrate only on Lung Patient, so it is the limitation of this study.
Author Response
≫We thank the reviewer for the kind words. Indeed, this study is only limited to lung patients but easily extensible to other cohorts.
Reviewer 3 Report
This submission is more suited to a more specialized journal like European Radiology, European Journal of Nuclear Medcine and Molecular Imaging, Radiology, among others. The readers of the Cancers maybe have limited interest in this piece.
Author Response
≫We thank the reviewer for the suggestion. We might consider other more clinical and more technical publications.
Reviewer 4 Report
Ivan Zhovannik et al. describe in their manuscript how the development of a prognostic model of disease outcome based on radiological images should be improved by considering variations in defining the tumor region in the image.
In the clinic defining the tumor region is a manual process and therefore inherently variable.
The influence of the variations in defining the tumor region were investigated using 5000 computer simulations per image on 420 clinical images of non-small cell lung cancer for training. The model generated parameters for 301 different values derived from the images to predict correctly the average survival time of the patient. Of the 5000 computer simulations of the computer model the one that produced over the 420 images the smallest variation in these parameters was selected. This computer model was used on some other 227 images to predict average survival time of the patients.
To test whether some parameters of the 301 are sensitive towards variations in defining the tumor region the authors compared the survival rate predictions using subsets of the 301 values used on 2 subsets of the 227 images. Some parameter subsets produced the same survival curves (robust values) on both image sets, other parameter subsets resulted in different survival curves. These are radiometric parameters which are sensitive to variations in the selection of the tumor region.
The conclusion of the authors is that when analysing the predictive power of a novel biomarker its sensitivity to variations in the determination of tumor region must be investigated as well.
Overall the article is very sound and the result is important for the evaluation of biomarkers in radiological image analysis. So, my recommendation is to publish the article with minor modifications.
I repeated the content of the article, as I understood it, in the above paragraph because the article could be a little bit clearer on the different groups of entities. There are three different groups, the images, divided in training (420) and evaluation set (227), the radiometric observable parameters per image (301) that are used in the model and the computer simulations (5000) with different tumor regions per image.
There are subsets of the evaluation set of the images (227) used and subsets of the parameters (301) to come to different survival curves in figure 6 and 7. I must admit that I am not completely certain that this interpretation of the article is correct. Hence, the use of groups and sub-groups is a little bit difficult to understand. Possibly the authors can be a bit clearer on the formation of groups defining them at the beginning in form of a legend and giving them symbolic names which are later used in brackets in the text.
The fact that the selection of the tumor region in the clinic is a manual process and inherently variable is the starting point of the investigation and belongs and the beginning of the article.
Author Response
Ivan Zhovannik et al. describe in their manuscript how the development of a prognostic model of disease outcome based on radiological images should be improved by considering variations in defining the tumor region in the image.
In the clinic defining the tumor region is a manual process and therefore inherently variable.
The influence of the variations in defining the tumor region were investigated using 5000 computer simulations per image on 420 clinical images of non-small cell lung cancer for training. The model generated parameters for 301 different values derived from the images to predict correctly the average survival time of the patient. Of the 5000 computer simulations of the computer model the one that produced over the 420 images the smallest variation in these parameters was selected. This computer model was used on some other 227 images to predict average survival time of the patients.
To test whether some parameters of the 301 are sensitive towards variations in defining the tumor region the authors compared the survival rate predictions using subsets of the 301 values used on 2 subsets of the 227 images. Some parameter subsets produced the same survival curves (robust values) on both image sets, other parameter subsets resulted in different survival curves. These are radiometric parameters which are sensitive to variations in the selection of the tumor region.
The conclusion of the authors is that when analysing the predictive power of a novel biomarker its sensitivity to variations in the determination of tumor region must be investigated as well.
Overall the article is very sound and the result is important for the evaluation of biomarkers in radiological image analysis. So, my recommendation is to publish the article with minor modifications.
≫ We would like to thank the reviewer for the kind words and the good description of our project. One minor addition: the ensembled model was selected based on the least squares of the difference between model coefficients and the median coefficient value, the median is derived based on the 5000 measurements.
I repeated the content of the article, as I understood it, in the above paragraph because the article could be a little bit clearer on the different groups of entities. There are three different groups, the images, divided in training (420) and evaluation set (227), the radiometric observable parameters per image (301) that are used in the model and the computer simulations (5000) with different tumor regions per image.
≫ That is a very precise description, indeed!
There are subsets of the evaluation set of the images (227) used and subsets of the parameters (301) to come to different survival curves in figure 6 and 7. I must admit that I am not completely certain that this interpretation of the article is correct. Hence, the use of groups and sub-groups is a little bit difficult to understand. Possibly the authors can be a bit clearer on the formation of groups defining them at the beginning in form of a legend and giving them symbolic names which are later used in brackets in the text.
≫ We agree with the reviewer that all the numbers make the text harder to interpret. In figure 6, we show that due to the uncertainty in the HIGH η group, there is a chance to get both significant and insignificant results; we then demonstrate the Kaplan-Meier curves just as a visualization of this concept.
≫ Our clinical reasoning behind the figure 7 was as follows. Imagine we let 301 clinicaians delineate the same gross tumor volume – will our model give the same prediction for those 301 delineations. The answer is – the less segmentation-induced uncertainty is against the effect, the more confident the model will be in its predictions.
The fact that the selection of the tumor region in the clinic is a manual process and inherently variable is the starting point of the investigation and belongs and the beginning of the article.
≫ We would like to express our gratitude to the reviewer for the important questions raised!